# The biophysical basis underlying the maintenance of early phase long-term potentiation

**Moritz F. P. Becker**[1]*, **Christian Tetzlaff**[1,2]*

**1** III. Institute of Physics – Biophysics, Georg-August University, Göttingen, Germany, **2** Bernstein Center for Computational Neuroscience, Göttingen, Germany

* moritz.becker@phys.uni-goettingen.de (MB); tetzlaff@phys.uni-goettingen.de (CT)

## Abstract

The maintenance of synaptic changes resulting from long-term potentiation (LTP) is essential for brain function such as memory and learning. Different LTP phases have been associated with diverse molecular processes and pathways, and the molecular underpinnings of LTP on the short, as well as long time scales, are well established. However, the principles on the intermediate time scale of 1-6 hours that mediate the early phase of LTP (E-LTP) remain elusive. We hypothesize that the interplay between specific features of postsynaptic receptor trafficking is responsible for sustaining synaptic changes during this LTP phase. We test this hypothesis by formalizing a biophysical model that integrates several experimentally-motivated mechanisms. The model captures a wide range of experimental findings and predicts that synaptic changes are preserved for hours when the receptor dynamics are shaped by the interplay of structural changes of the spine in conjunction with increased trafficking from recycling endosomes and the cooperative binding of receptors. Furthermore, our model provides several predictions to verify our findings experimentally.

**Data Availability Statement:** All relevant data are within the manuscript and its Supporting information files.

## Author Summary

The cognitive ability of learning is associated with plasticity-induced changes in synaptic transmission efficacy mediated by AMPA receptors. Synaptic changes depend on a multitude of molecular and physiological mechanisms, building complex interaction networks. By formalizing and employing a biophysical model of AMPAR trafficking, we unravel and evaluate the interplay between key mechanisms such as receptor binding, exocytosis, morphological changes, and cooperative receptor binding. Our findings indicate that cooperative receptor binding in conjunction with morphological changes of the spine and increased trafficking from recycling endosomes leads to the maintenance of synaptic changes on behaviorally relevant time spans.

**Funding:** This work has been funded by the German Research Foundation (CRC1286, project C1 and project ⩝#419866478) (to CT). Funder website: https://www.sfb1286.de/. The funders had no role in study design, data collection and analysis, decision to publish, or preparation of the manuscript.

**Competing interests:** The authors have declared that no competing interests exist.

## Introduction

Synaptic plasticity regulates synaptic transmission strength and is associated with cognitive processes such as learning and memory. For excitatory synapses, the transmission strength is mainly determined by the number of $\alpha$-amino-3-hydroxy-5-methyl-4-isoxazolepropionic acid receptors (AMPARs) accumulating at the postsynaptic density (PSD), which is a dense, membrane-less organelle located within dendritic spines [1–4]. Dendritic spines are membraneous protrusions that serve as compartments for proteins and organelles involved in synaptic transmission. Long-term potentiation of synaptic strength (LTP) can be divided into different phases such as early-phase LTP (E-LTP), which decays by 1-6 hrs after induction, or late-phase LTP (L-LTP), which lasts for several hours or even days and depends on the synthesis of new proteins [5–7]. While reports have identified various molecular processes and pathways underlying the different phases of LTP, especially for L-LTP [6, 8, 9], the biological and biophysical mechanisms maintaining E-LTP remain unclear. To address this issue, here we focus on protein synthesis independent E-LTP and investigate the influence of different experimentally motivated candidate mechanisms on AMPAR dynamics at the PSD [10].

AMPARs are continuously inserted and removed from the spine membrane by exocytosis and endocytosis [11–14]. Experimental studies indicate that E-LTP decays after several minutes if the exocytosis of additional AMPARs is blocked ([13, 15–17]; Fig 1A, triangles).

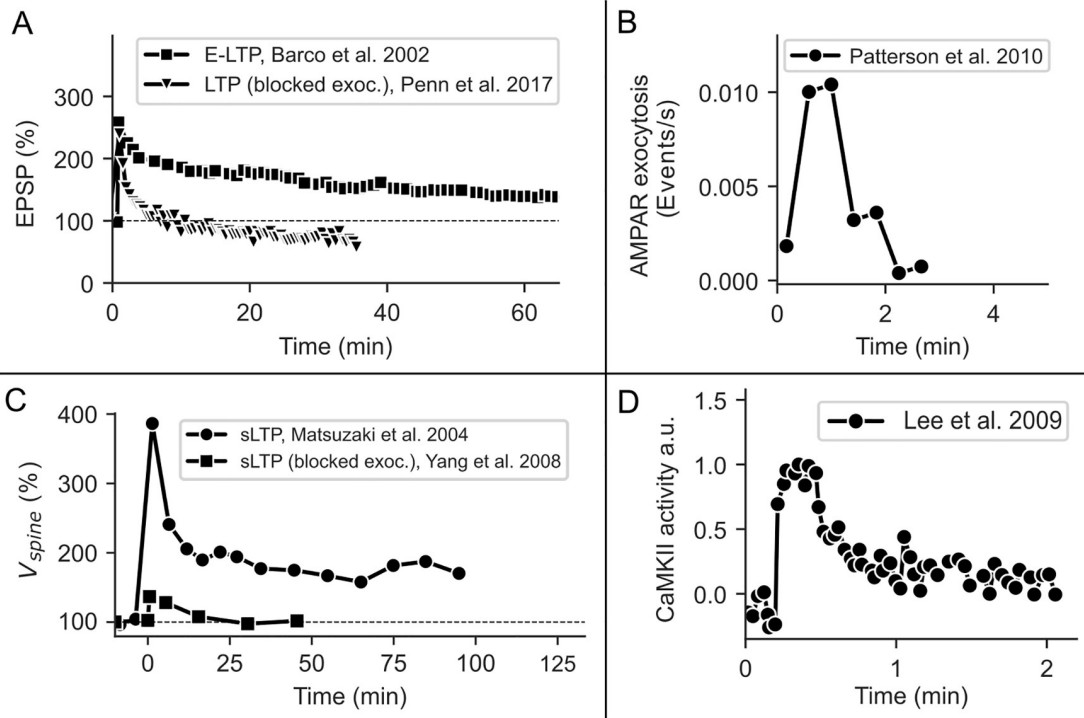

**Fig 1. Temporal course of E-LTP and related processes. (A)** E-LTP decays after 1-6 hours (squares; data taken from [10]), while L-LTP remains stable for hours to days ([6]; not depicted here). When exocytosis is blocked at the postsynaptic spine, E-LTP decays after 10-20 min (triangles; data taken from [15]). **(B)** The exocytosis event rate of GluA1 at spines is increased for a few minutes after LTP-induction (data taken from [12]). **(C)** Functional LTP is accompanied by structural changes of the spine, which also depends on exocytosis (data taken from [19, 20]). **(D)** The activity of CaMKII is also increased for a few minutes after LTP-induction (data taken from [21]). The figures have been redrawn from [10] Figure 3C, [15] Figure 2C, [12] Figure 4A, [19] Figure 2C, [20] Figure 4C, [21] Figure 2B.

Although these results suggest that exocytosis could be the main mechanism underlying the maintenance of E-LTP, the rate of exocytosis events at dendritic spines is only increased for a few minutes after LTP-induction and exocytosis of receptors is slow compared to diffusional exchange ([11, 12, 18]; Fig 1B).

AMPARs are regularly exchanged between the postsynaptic and the dendritic membrane via lateral diffusion but can become bound and thereby functional at the PSD [14, 22–24]. Several studies indicate that the phosphorylation and interaction of PSD scaffolding proteins such as PSD95 and transmembrane AMPAR regulatory proteins (TARPs) such as Stargazin play a key role in binding AMPARs at the synapse [3, 25, 26]. However, the activity of kinase CaMKII which targets these PSD proteins is only upregulated for a few minutes after LTP-induction ([21]; Fig 1C). While the phosphorylated state of the TARPS probably outlasts CaMKII activity, the fast decay of LTP in the case of blocked exocytosis indicates that AMPAR binding rates are not increased for much longer [15, 21]. Also, concentrations of PSD proteins only increase in the presence of protein synthesis and persistent spine enlargement, indicating that the number of possible binding sites stays approximately constant during E-LTP [27, 28]. Taken together, these results demonstrate that the principles underlying the maintenance of E-LTP remain unknown as neither the dynamics of the exocytosis rate nor the CaMKII activity seems to match the slow decay of E-LTP (Fig 1A, squares).

However, additional biological and biophysical mechanisms influence the AMPAR dynamics at the dendritic spine. The spine itself is not static but shaped by the dynamics of the spine's cytoskeleton. The induction of LTP also yields structural modifications of the spine (sLTP), which is expressed by an increase in spine volume due to actin polymerization, and the translocation of AMPAR-containing recycling endosomes (REs) into the spine [19, 20, 27, 29–34] (Fig 1C). Both could influence AMPAR trafficking: RE translocation by increasing receptor recycling to the surface via AMPAR exocytosis [12, 29, 30] and spine growth by regulating diffusion between dendrite and spine [35–37]. Exocytosis from REs not only delivers AMPARs to the spine membrane but also stabilizes the spine after LTP induction and has been suggested to be involved in the early maintenance of LTP [20, 29–31, 38]. Importantly, changes in the spine volume and RE allocation last much longer than changes in the rate of exocytosis events at the spine [12, 18, 27, 29].

Furthermore, AMPARs are not homogeneously distributed in the PSD but cluster on PSD nanodomains formed by PSD95 and other scaffolding proteins [39, 40]. The emergence of such clusters can be related to receptor-receptor and receptor-scaffolding protein interactions, which in turn influence the binding of AMPARs [41–45]. The relationship between structural plasticity, AMPAR clustering, and the induction and maintenance of E-LTP remains unexplained.

In this study, based on well-established theoretical studies [45, 46] we derive a mathematical model of AMPAR trafficking in a dendritic spine that integrates diverse experimental data [12, 19–21, 24, 29, 47–51]. We use this model to systematically investigate the impact of the above-mentioned biological mechanisms on the maintenance of E-LTP by comparing the model with the experimental data (summary of experimental findings shown in Fig 1, data taken from [10, 12, 15, 21]).

We find that morphological changes in conjunction with increased RE trafficking and cooperative binding can complement each other and together provide a possible explanation for the decay time of E-LTP as well as the dependence of LTP on exocytosis. This combined model provides a new hypothesis about the principles underlying the maintenance of E-LTP and leads to specific, new experimental predictions that could verify our findings.

## Results

### Basic model of AMPAR-trafficking

The basic model integrates the core processes of AMPAR trafficking in the dendritic spine: lateral diffusion to and from the dendritic membrane, exo- and endocytosis within the spine, and binding to and disassociation from scaffolding proteins in the PSD (Fig 2A). We formalize the coaction of these processes by considering the number of AMPARs that freely move on the membrane of the spine denoted by $U$ and of AMPARs that are bound in the PSD labeled by $B$. The surface area of the spine is denoted by $A_{spine}$. Please note that we do not explicitly model the number of AMPARs in the intracellular space of the spine or on the dendritic membrane. As the rate of exocytosis is slow [11, 12] and diffusion of surface receptors is fast [23] we assume that the dendritic and intracellular pools are rapidly replenished. On the synaptic membrane, $U$ is increased by exocytosis at a rate $k_{exo} S_{exo}$ where $S_{exo}$ denotes the size of an exocytic event and $k_{exo}$ the rate at which exocytosis events occur. $U$ is further increased by lateral diffusion from the dendritic to the spine membrane at a rate $k_{in}$, and by the release of AMPARs from the PSD described by $k_{BU} B$. $U$ is decreased by endocytosis at a rate $k_{endo} \frac{U}{A_{spine}}$ and lateral diffusion to the dendrite at a rate $k_{out} \frac{U}{A_{spine}}$, as well as by AMPARs binding to the PSD scaffold described by $k_{UB}(P - B) \frac{U}{A_{spine}}$, where $P$ denotes the number of slots formed by scaffolding proteins. Thus, the change in the number of unbound AMPARs can be formalized by

$$\frac{dU}{dt} = k_{exo} S_{exo} + k_{in} + k_{BU} B - (k_{endo} + k_{out} + k_{UB}(P - B)) \frac{U}{A_{spine}}. \qquad (1)$$

We do not consider direct endo- and exocytosis of bound AMPARs such that the number is only regulated by the interchange with freely moving AMPARs.

$$\frac{dB}{dt} = k_{UB}(P - B) \frac{U}{A_{spine}} - k_{BU} B. \qquad (2)$$

Eqs (1) and (2) describe the AMPAR-dynamics on the spine's membrane under basal condition. We derived the corresponding set of parameter values of an average spine by estimating the values from several experimental studies that are mainly conducted in hippocampal cultures and slices (see Table 1) and are described in more detail in the methods section. Note that the receptor flux, for example for diffusion out of the spine or for receptor binding, depends on the receptor concentration $U/A_{spine}$. Thereby, some of the rate constants have units of $\mu m^2/s$. This follows from the requirement for equal units on the left and right hand site of Eqs (1) and (2).

### Exocytosis event and binding rate increase alone cannot account for the maintenance of E-LTP

To examine whether a brief increase in exocytosis event and binding rate can account for the maintenance of E-LTP for several hours as well as the fast decay in the absence of exocytosis, we first study a basic model of AMPAR trafficking. Note that we compare bound AMPARs to EPSP data, which is given in percentage of the baseline value. Also, many experimental methods do not yield quantitative data on receptor numbers, e.g. fluorescence microscopy. Therefore, in this study, receptor numbers are shown in percentage of the baseline level, if not stated otherwise.

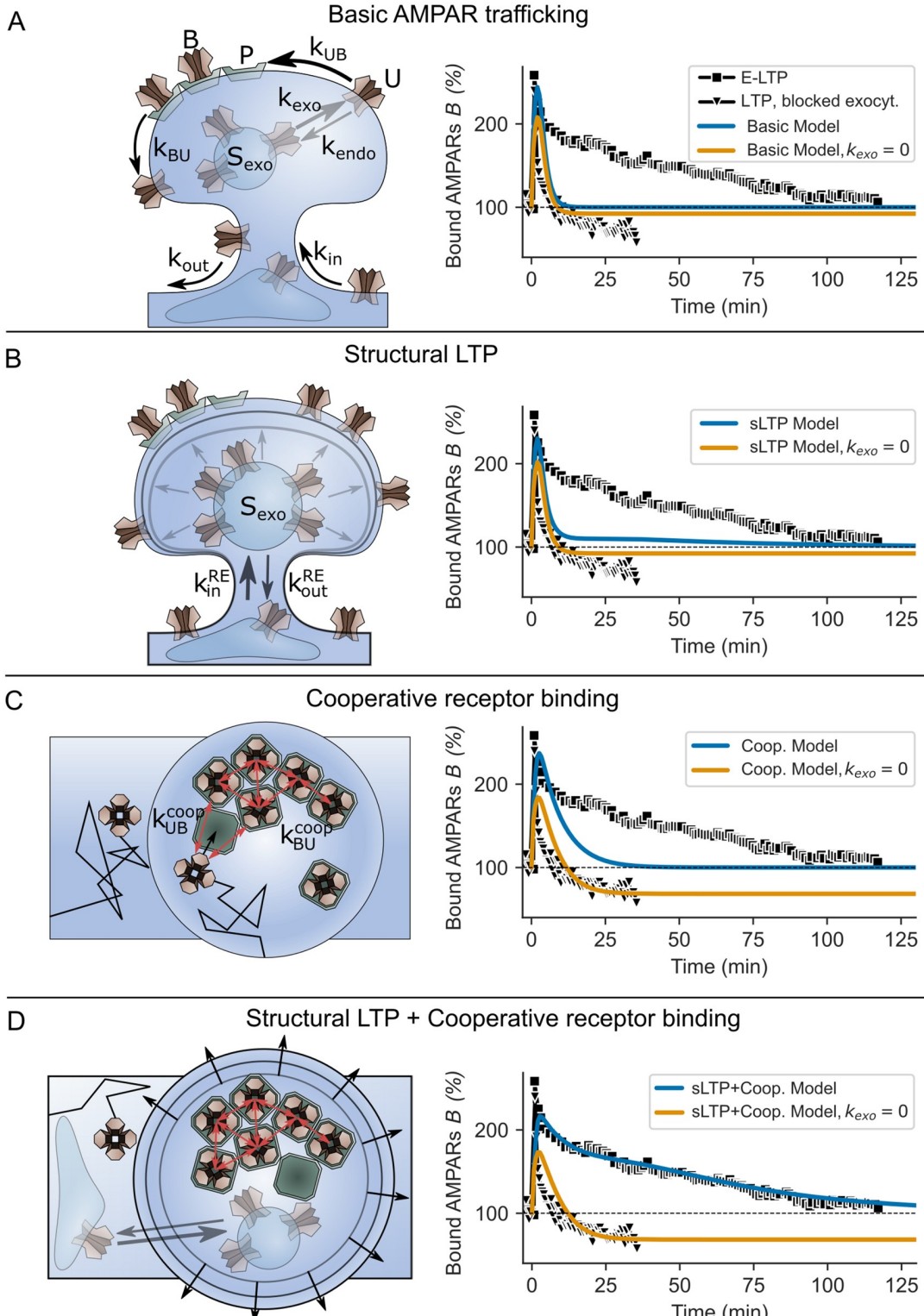

**Fig 2. Different model versions and resulting AMPAR dynamics compared to experimental data. (A, left)** *Basic model*: AMPA receptors are trafficked across the spine via several pathways. AMPARs are transported from recycling endosomes to the membrane via exocytosis. During this process that occurs at a rate $k_{exo}$, a number of receptors $S_{exo}$ enters the spine membrane. Similar, AMPARs are inserted into the spine via endocytosis at a rate $k_{endo}$. In addition, mobile receptors $U$ can laterally diffuse into and out of the spine at rates $k_{in}$ and $k_{out}$. However, AMPARs only become functional if they are bound to

PSD binding sites $P$. Binding and unbinding occurs at rates $k_{UB}$ and $k_{BU}$. **(right)** During LTP-induction, exocytosis ($k_{exo}$) and binding rate ($k_{UB}$) are briefly increased. The LTP-induced dynamics of the basic model (blue line) is compared to experimental data of E-LTP (black squares; data taken from [10]) and also to LTP with the blockage of exocytosis (orange line compared to black triangles; data taken from [15]). **(B, left)** *sLTP model*: We assume that the number of AMPARs that enter the membrane during an exocytosis event $S_{exo}$ depends on the spine volume and RE trafficking, both of which change during LTP [19, 29]. Therefore, sLTP influences the LTP-induced AMPAR dynamics in addition to $k_{exo}$ and $k_{UB}$. **(right)** Similar to (A), $B$ increases during LTP induction but only slowly decays back to the baseline if exocytosis occurs. **(C, left)** *Cooperative model*: We hypothesize that binding of receptors is cooperative (red arrows depict receptor-receptor interactions). **(right)** Cooperative receptor binding slows down the decay of LTP induced increase in bound AMPARs and increases the sensitivity to exocytosis. **(D, left and right)** *sLTP+Cooperative Model*: sLTP and cooperative receptor binding in concert with transiently increased exocytosis event and binding rate sustain E-LTP. **(A-D)** LTP-induction is triggered at time point $0s$.

We model the LTP-induced variation of the exocytosis event and binding rate (Fig 1B and 1D). Parameter values under basal conditions, i.e. before LTP induction, will be denoted by a superscript index 0 ($k_{exo}^0$, $k_{UB}^0$). The time-dependence of $k_{exo}$ and $k_{UB}$ is described by an exponential rise ($\tau_{exo}^1$, $\tau_{UB}^1$) and decay ($\tau_{exo}^2$, $\tau_{UB}^2$) with maximum amplitudes of $A_{exo}$ and $A_{UB}$ (see Table 2 and Eqs (6)–(8)). Whereas the rates $k_{out}$ and $k_{endo}$ are kept constant, the total efflux via the corresponding trafficking pathways does vary in response to receptor concentration changes ($U/A_{spine}$) (see Eq (1)). Please note that the extensions of the basic model to also include structural plasticity or cooperative receptor binding are described in the corresponding sections below.

Under basal conditions, the different processes of AMPAR-dynamics described before balance each other such that the number of mobile and bound AMPARs in the PSD and on the

**Table 1. Parameters of AMPAR trafficking.**

| Parameter | Meaning | Value | Experiments |
|---|---|---|---|
| $k_{exo}^0$ | Rate of exocytosis events at the spine under basal condition | $0.0018\ s^{-1}$ | [12] |
| $k_{endo}$ | Rate of receptor endocytosis at the spine | $0.0021\ \mu m^2/s$ | This study, Eq (4) |
| $k_{out}$ | Rate at which AMPARs exit the spine via lateral diffusion | $0.018 \mu m^2/s$ | [22, 51], this study |
| $k_{in}$ | Rate at which AMPARs enter the spine via lateral diffusion | $0.2\ \#/s$ | [12, 35, 49, 50] |
| $k_{BU}$ | Unbinding rate of receptors from PSD slots | $0.1\ s^{-1}$ | [24] |
| $k_{UB}^0$ | Binding rate of AMPARs at the PSD | $0.0036\ \mu m^2/(\#s)$ | This study, Eq (5) |
| $k_{UB}^0$ (Coop.) | Binding rate of AMPARs at the PSD in the cooperative binding model | $0.0005\ \mu m^2/(\#s)$ | This study, Eq 23 |
| $U^*$, $B^*$ | Steady state number of mobile and bound receptors at the average spine | 10, 20 # | [48, 52] |
| $P$ | Number of slots at the PSD | 70 # | This study |
| $V_{spine}^0$ | Spine volume | $0.08\ \mu m^3$ | [19, 37] |
| $S_{exo}^0$ | Size of an AMPAR exocytosis event | 13 # | [47] |

**Table 2. Parameters of LTP-induced changes in exocytosis event and binding rate.**

| Parameter | Meaning | Value | Experiments |
|---|---|---|---|
| $A_{UB}$ | Factor of LTP-induced increase in binding rate in the basic model | 30 | This study |
| $A_{UB}$ (Coop.) | Factor by which the binding rate increases upon LTP-induction in the cooperative binding model | 5 | This study |
| $\tau_{UB}^1$ | Time constant of binding rate increase during LTP induction | $5\ s$ | [15, 21] |
| $\tau_{UB}^2$ | Decay time constant of binding rate after LTP induction | $60\ s$ | [15, 21] |
| $A_{exo}$ | Factor by which the exocytosis event rate increases upon LTP-induction | 5 | [12, 13] |
| $\tau_{exo}^1$ | Time constant of exocytosis event rate increase during LTP induction | $25\ s$ | [12] |
| $\tau_{exo}^2$ | Decay time constant of exocytosis event rate after LTP induction | $60\ s$ | [12] |

synaptic membrane reach an equilibrium state of constant temporal averages with 28.5% of slots being occupied (parameters have been chosen explicitly to result in a low occupancy of the PSD; see Methods).

The induction of LTP and the related increase in receptor exocytosis and binding lead to a fast increase in the number of mobile and bound AMPARs, which results in a stronger synaptic transmission strength, assuming that AMPAR EPSCs are proportional to the number of bound AMPARs at the PSD (Fig 2A, blue line). After a few minutes, the exocytosis event and binding rate return to basal levels. Given these dynamics, the increased level of bound AMPARs in the PSD declines such that it reaches the pre-LTP level after about 10–15 minutes. The time course of bound AMPARs is similar in the case of blocked exocytosis (Fig 2A, orange line). This result indicates that some core mechanisms are missing, which maintain bound AMPARs in the PSD for a longer duration.

## Structural plasticity is not sufficient to explain the maintenance of E-LTP

The previous implementation of the model does not account for structural spine changes and the accompanied mobilization of recycling endosomes [19, 29, 30]. We hypothesize that, whereas the rate of exocytosis events at the spine only increases transiently (Fig 1B, [12]), the rise in recycling endosome availability during the course of sLTP results in enhanced receptor recycling and thereby in the growth of the receptor exocytosis event size $S_{exo}$. These assumptions are supported by experimental data showing that spine growth, membrane trafficking from recycling endosomes and AMPAR insertion are tightly coupled, all increasing about two-fold during LTP [29, 30]. We constructed a reduced model in which spine volume changes ($V_{spine}$) are followed by a proportional but delayed change in the exocytosis event size $S_{exo}$ (Eqs 12 and 13, Fig 3C). Variations in spine volume and surface area also influence the average dwell time of receptors on the spine membrane, the concentration of mobile receptors on short time scales (below equilibration time) and the total receptor surface concentration ($(U + B)/A_{spine}$). Therefore, the model composes a change in exocytosis and spine area (Eqs 1 and 2). In addition, the dependence of sLTP on the initial spine size is accounted for [19] (Fig 3C), whose influence on E-LTP expression will be investigated later on (for model details see Materials and methods).

All parameters of sLTP are based on previous studies ([19, 20, 27, 37]; see Materials and methods). Similar to the basic model, the induction of LTP leads to a strong and fast increase in the number of bound AMPARs; however, the level of bound AMPARs falls significantly slower with sLTP than in the basic model (Fig 2B). This slower decay can be attributed to the increase in mobile receptor concentration that is mediated by the change in $S_{exo}$ (Fig 3C and 3D) and disappears if exocytosis is blocked (Fig 2B, orange line). Given normal conditions, the time span until the AMPAR level falls back to the basal level is similar to experiments ($\approx 100$ *min*). However, between 10 and 100 minutes the overall level of bound AMPARs is lower than in experimental data. Thus, the influence of the sLTP on the AMPAR-dynamics is not sufficient to explain the maintenance of E-LTP.

## Cooperative receptor binding is not sufficient to explain the maintenance of E-LTP

Our model of cooperative AMPAR-dynamics, based on [45], enables us to investigate the influence of cooperative binding effects on the maintenance of E-LTP. In this model, the probability of receptors to bind to or unbind from a slot depends on the number of nearest neighbours on a grid that represents the PSD (Fig 3A). We used a square grid, such that the number of nearest neighbours is at maximum 8. We developed a mean-field approximation

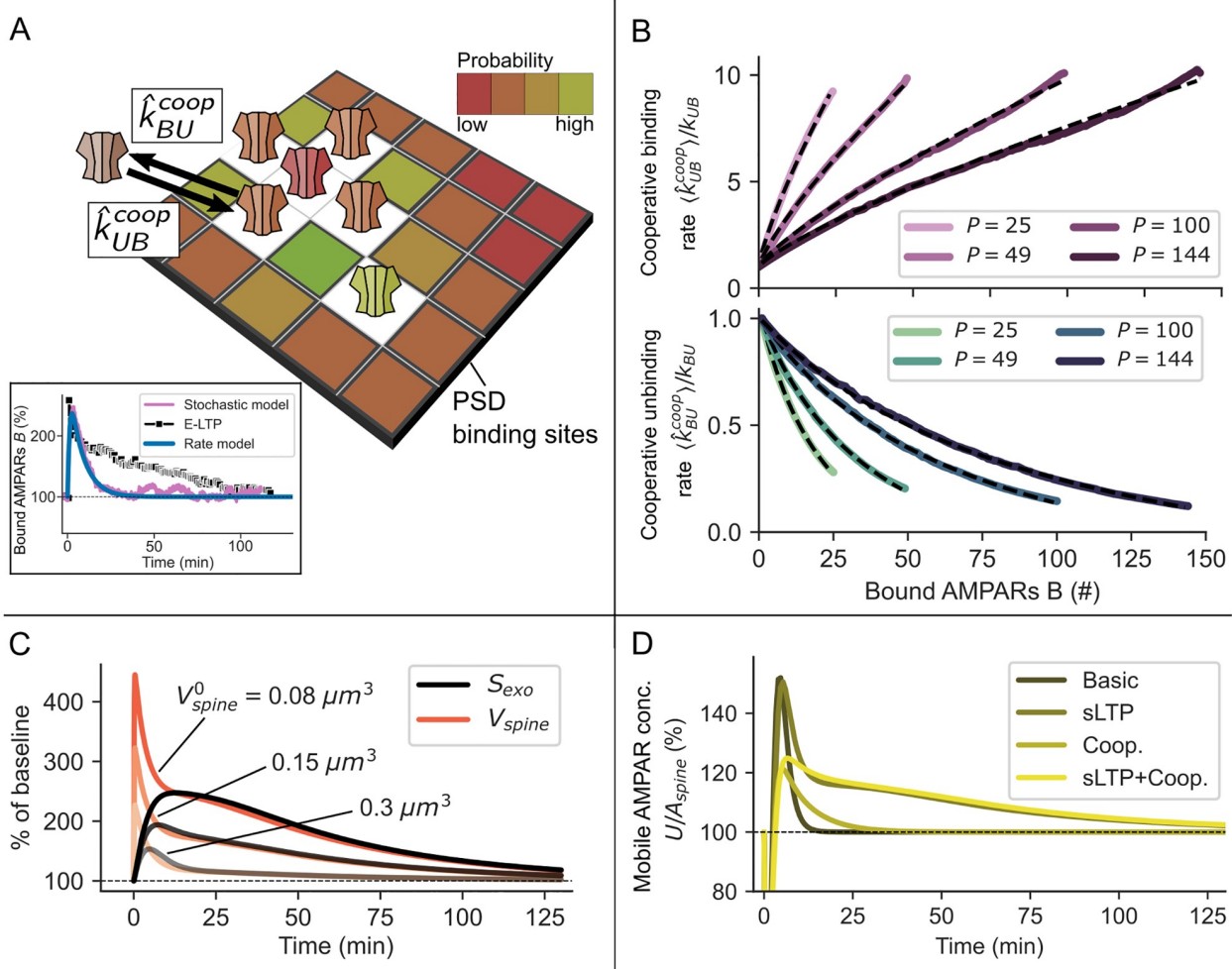

**Fig 3. Basic principles and results of cooperative receptor binding and sLTP. (A)** Sketch of the cooperative model mechanisms: Slots of the PSD are spatially ordered into a grid with each slot having a binding $\hat{k}_{UB}^{coop}$ and unbinding $\hat{k}_{BU}^{coop}$ rate dependent on the number of nearest neighbours, indicated by different colors (white: occupied slot). **(B)** The mean binding $\langle\hat{k}_{UB}^{coop}\rangle$ and unbinding rate $\langle\hat{k}_{BU}^{coop}\rangle$ (Eqs (16) and (17)) vary with the grid size (slots $P$) and the total number of bound receptors. The simulation results are fitted by Eqs (19) and (18) (dashed lines). The dynamics of the resulting mean-field model match well with the stochastic simulations (**A**, inset). **(C)** The enlargement of the spine during sLTP is accompanied by an increase in the size of exocytosis events $S_{exo}$ (Eqs (13) and (12)) and depends on the initial size of the spine $V_{spine}^0$. **(D)** The concentration of mobile receptors $U/A_{spine}$ increases after LTP-induction but is only sustained if accompanied by sLTP.

of this stochastic grid model (Eqs (18) and (19)), which is integrated into the basic model (Eqs (1) and (2)). In the mean-field model, cooperativity results in effective binding and unbinding rates $k_{UB}^{coop}$, $k_{BU}^{coop}$ that depend on the number of already bound receptors $B$ as well as the number of binding sites $P$ (Fig 3B). Without loss of generality, we assume that the synapse contains a single domain of scaffolding proteins; therefore, P represents the total number of binding sites at the synapse. Like before, LTP-related changes of the exocytosis event and binding rate are modeled by temporal variations of $k_{exo}$ and $k_{UB}$. Similar to the basic model, LTP-induction yields a fast increase in the number of bound AMPARs followed by a fast decay (Fig 2C, blue line). Compared to the basic model, the decay is slower such that the level of bound AMPARs reaches the basal condition after ≈30–35 minutes; however, it is significantly faster than in experiments. The blockage of exocytosis, after LTP-induction, reduces the level of bound AMPARs to ≈80% (orange line), which is different from the basic or sLTP

model. This effect of reduced transmission strength is also found in experiments (Fig 2C, triangles). Thus, the cooperative effect between receptors is not sufficient to explain the maintenance of E-LTP.

## Exocytosis is the limiting factor in E-LTP maintenance

Although based on experimental data, the parameters considered before could embrace an imprecision in their exact values as exocytosis might be underestimated in experiments [53] and the amplitude and decay of the binding rate during LTP is uncertain. We, therefore, utilize a random sampling method (see Materials and methods) to find the parameter set that results in the best match of the AMPAR-dynamics of the model with the experimentally measured temporal course of transmission strength (Fig 1A). For this, we randomly drew 2000 different parameter values for the trafficking rates $k_{BU}, k_{UB}^0, k_{exo}^0, k_{endo}$ and the LTP-induced temporal development of exocytosis ($\tau_{exo}^2, A_{exo}$) and receptor binding rates ($\tau_{UB}^2, A_{UB}$) from a reasonable regime (see Table 3). We simulated the resulting AMPAR-dynamics for the basic model and its extensions by sLTP or cooperative binding discussed before. To find the best match, for each trial we use a distance measure (Eq (9)) between the temporal development of the number of bound AMPARs in the model and the experimentally measured development of the EPSP with and without blockage of exocytosis (see Materials and methods).

Indeed, for all three model versions, several parameter sets yield dynamics that are in agreement with experimental findings with and without the blockage of exocytosis (Fig 4A, colored lines; the 0.5% best matching parameter sets are shown). Furthermore, the temporal development of the binding rate is consistent with the CaMKII activity during LTP-induction (Fig 4B). However, for the basic model and the sLTP-model, the exocytosis event rate becomes much too high compared to experiments and the decrease becomes too slow (Fig 4C). Thereby, mobile receptor surface concentration $U/A_{spine}$ increases by about 200%, which is far beyond experimental observations ($\approx$20% for GluA1, [27]). For the model of AMPAR-dynamics with cooperative binding, the exocytosis event rate stays in a reasonable regime but also decays too slowly (lime green). However, the latter result indicates that the increase in exocytosis due to sLTP might be sufficient to maintain E-LTP in the cooperative binding model. This is also apparent from the increase in the mobile receptor concentration necessary to mediate E-LTP in the cooperative model (Fig 4D), which is close to the increase during sLTP ($\approx$25%, Fig 3D). We repeated this procedure for different values of the diffusion parameter $k_{in}$ and $k_{out}$ and the steady state $U^*, B^*$, yielding to similar results (S1 Fig).

**Table 3. Random parameter sampling.**

| Parameter | Meaning | Range |
|---|---|---|
| $k_{exo}^0$ | Rate of exocytosis events at the spine under basal conditions. | $0\ldots0.01$ $s^{-1}$ |
| $\tau_{exo}^2$ | Decay time constant of exocytosis event rate after LTP induction. | $0\ldots67$ *min.* |
| $A_{exo}$ | Factor by which the exocytosis event rate increases upon LTP-induction. | $0\ldots20$ |
| $k_{BU}$ | Unbinding rate of AMPARs from slots at the PSD. | $0\ldots1\ s^{-1}$ |
| $\tau_{UB}^2$ | Decay time constant of binding rate after LTP induction. | $0\ldots8$ *min.* |
| $A_{UB}$ | Factor by which the binding rate increases upon LTP-induction. | $0\ldots50$ |
| $A_{UB}(Coop.)$ | Factor by which the binding rate increases upon LTP-induction in the cooperative binding model. | $0\ldots20$ |

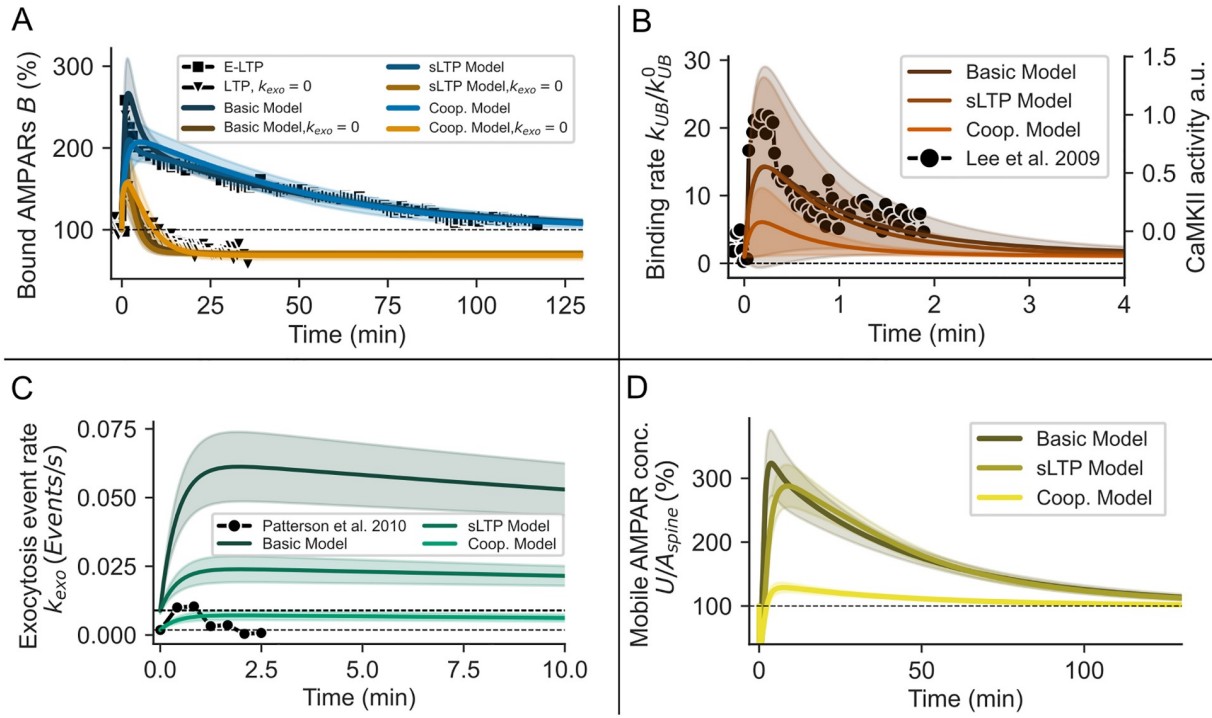

**Fig 4. Variation of parameter values yields parameter sets for all three model versions such that E-LTP is maintained for several hours.**
**(A)** The experimentally measured temporal course of E-LTP with and without the blockage of exocytosis [10, 15] is matched by all three model versions (basic, sLTP, cooperative) with corresponding parameter sets. Please note that lines overlie each other. **(B,C)** For the parameter sets used in (A), the LTP-induced increase of the binding rate of the models matches experimental data of CaMKII [21] (B), whereas the exocytosis event rate is too high for the basic and sLTP model and decays too slow for all three model versions compared to experimental data [12] (C). **(D)** The concentration of mobile receptors $U$ increases to ≈300% of the baseline level for the basic and the sLTP model but only to ≈130% for cooperative model. **(A-D)** Shown are the upper 0.5% of 2000 randomly sampled parameter sets that match best with the experimental data in (A). For the matching we used the cost function described by Eq (9). Shades indicate standard deviation across parameter sets.

## sLTP and cooperative binding complement each other to maintain synaptic potentiation

We have shown that neither the basic AMPAR dynamics nor its extension by sLTP or cooperative binding seems to explain the maintenance of E-LTP. Next, we test whether a combination of cooperative binding and sLTP complement each other to mediate E-LTP as observed experimentally. All model parameters and the temporal developments of exocytosis event rate, binding rate, and spine volume are the same as before. In this combined model with experimentally derived parameters, the number of bound AMPARs increases rapidly after LTP induction (Fig 2D). If exocytosis is blocked, this increase is followed by a rapid decay below the baseline level (orange line). In presence of exocytosis, however, the combined dynamics of sLTP and cooperative binding sustain the level of bound AMPARs at a level that matches experimental data (blue line). But why are both mechanisms required to maintain E-LTP?

To provide an answer, we assume that the trafficking dynamics of AMPARs are much faster than the changes triggered by LTP. This allows us to consider the fixed point of bound AMPARs $B^*$ as a function of the spine volume $V_{spine}$ and respectively the size of exocytosis events $S_{exo}$ (Fig 5A). This function illustrates the different influences cooperative binding, morphological changes, and RE trafficking have on the number of bound AMPARs:

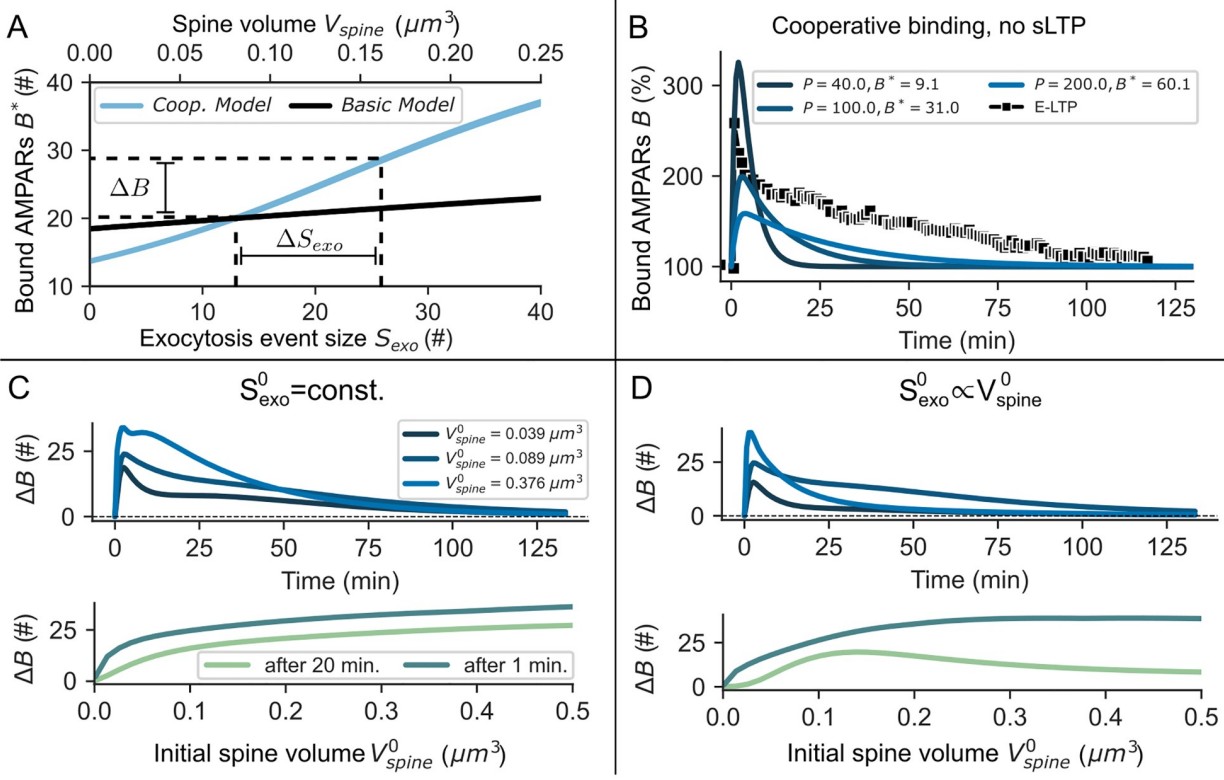

**Fig 5. The decay of E-LTP is determined by the size of spine and PSD. (A)** In the cooperative model (blue line), the number of bound AMPARs in the fixed point $B^*$ strongly depends on the amount of exocytosis from REs ($S_{exo}$) compared to the basic model (black line). Thereby, small changes in the amount of exocytosed AMPARs $\Delta S_{exo}$ are sufficient for LTP expression ($\Delta B$). **(B)** The decay time and relative amplitude of LTP change with the size of the PSD $P$. Whereas bound AMPARs $B$ show fast dynamics for small PSDs (dark blue line), $B$ only slowly decays back to the steady state baseline for large PSDs (light blue line). **(C)** E-LTP for different spine sizes where the PSD area scales linear with the spine surface area ($P \propto A_{spine}$) but the amount of baseline AMPAR exocytosis is kept constant ($S^0_{exo} = 13$). Top: Although large spines lack pronounced sLTP, the slower dynamics due to the larger PSD result in E-LTP with a time scale similar to that of smaller spines. Bottom: The change in AMPAR abundance at the PSD 1 and 20 min. after LTP induction. **(D)** Same as in (C) but $S^0_{exo}$ depends on the initial spine volume $V^0_{spine}$ according to Eq (13). Top: E-LTP decays rapidly for large and very small spines. Bottom: Same as in (C).

- *Cooperative binding* influences the shape of this function. If the AMPAR dynamics are determined by cooperative binding (Fig 5A, blue line), the function becomes steep and sigmoidal-shaped. This is significantly different compared to the function of the basic model, which has a shallow slope (Fig 5A, black line).

- *Morphological changes* influence the argument of the function. A variation of the spine volume and the exocytosis event size shifts the system to a new state with a different number of bound AMPARs.

Thus, AMPAR-trafficking dynamics only extended by sLTP (black line) would imply that an increase by $\Delta S_{exo}$ results in a small change of the number of bound AMPARs. By contrast, if we additionally consider cooperative binding between AMPARs (blue line), a shift in $V_{spine}$ and therefore in $S_{exo}$ results in a significant increase of the number of bound receptors (here, $\Delta B$).

## Decay of E-LTP depends on PSD and spine size

For the results discussed before, we considered an average spine with a spine volume of $V^0_{spine} = 0.08 \mu m^3$ before LTP-induction and a constant PSD size with 70 binding sites. Next, we evaluate to what degree E-LTP maintenance depends on the initial volume $V^0_{spine}$ with the corresponding number of slots $P$. We assume that the number of slots $P$ changes linearly with the initial surface area of the spine $A^0_{spine}$:

$$P = P^{basal} \cdot \frac{A^0_{spine}}{A^{basal}_{spine}}, \tag{3}$$

where $P^{basal}$ = 70 and $A^{basal}_{spine} = 0.898 \mu m^2$ such that the parameter values used before still hold (Table 1). The surface area is calculated from the volume assuming spherical spines. In addition, experimental studies have shown that sLTP mainly occurs for small to intermediate spines but not for large spines with volumes above about $0.3 \mu m^3$ [19, 34]. We integrated this dependence of $\Delta V_{spine}$ on $V^0_{spine}$ into our model using data from several studies [19, 20, 34] (Eq (11)).

First, we assume that the size of exocytosis events under basal conditions does not depend on the initial spine volume but is constant ($S^0_{exo} = 13$). Surprisingly, in this case, E-LTP is sustained even for large spines where sLTP is not as pronounced (Fig 5C). Similarly, we find that the system takes longer to return to equilibrium after LTP-induction as we increase the PSD size $P$ while keeping $V_{spine}$ constant and independent of $P$ (Fig 5B). This effect is also present in the non-cooperative model but much less pronounced (not shown). This shows that spines with large PSDs, having more binding sites, are much more efficient in keeping newly inserted receptors if binding is cooperative. We find three indications for why this is the case. First, during LTP induction, large PSDs are less occupied than small PSDs, which promotes rebinding ($max(B)/P \approx 68\%$ for P = 40, $\approx 62\%$ for P = 100 and $\approx 50\%$ for P = 200, Fig 5B). Second, larger systems take longer to equilibrate in general. Third, even though a sigmoidal dependence between bound and mobile AMPARs is present for different PSD sizes, it is more pronounced for large PSDs (Fig 6A). Thereby, for the latter, a change in the mobile receptor concentration has a stronger impact on the number of bound receptors as compared to small PSDs.

If the initial size of exocytosis events $S^0_{exo}$ scales with the initial spine volume (Eq (13)), we find that E-LTP is sustained for intermediate initial spine sizes $V^0_{spine} \approx 0.1 \mu m^3$ (Fig 5D). For very small ($V^0_{spine} < 0.03 \mu m^3$) and large spines ($V^0_{spine} > 0.35 \mu m^3$), however, E-LTP decays after about 25 minutes, matching experimental insights indicating that large spines do not experience long-lasting LTP [19, 34]. We argue that if exocytosis increases with the spine size, the PSD will saturate due to the larger influx of receptors. Thereby, receptors that are added to the spine during LTP induction only have a low chance of getting bound such that $B$ quickly returns to the baseline in the absence of sLTP.

## Model predictions

This study yields some experimentally verifiable predictions. Our main prediction is the requirement of the interplay between cooperative binding of receptors and morphological spine changes accompanied by an increase in AMPAR trafficking to the spine surface from REs during E-LTP (Fig 2D). By contrast, each mechanism alone does not lead to the required accumulation of bound receptors for a time span of several hours (Fig 2A–2C). We predict that due to cooperative effects, increased exocytic trafficking of AMPARs is already sufficient for synaptic potentiation without the need for additional factors that stabilize AMPARs at the

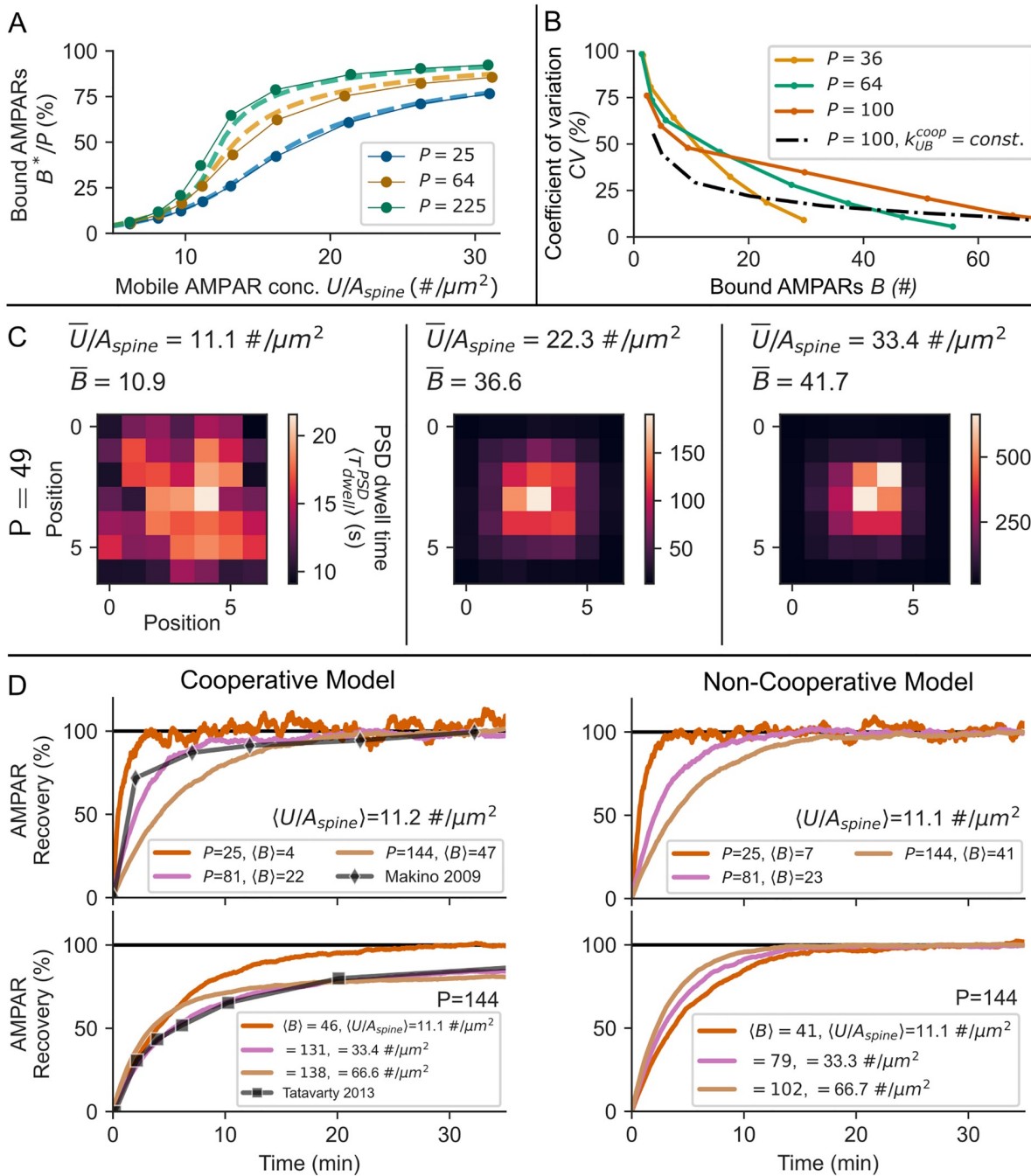

**Fig 6. Receptor clustering influences the statistical properties and mobility of receptors at the PSD. (A)** Characteristic for cooperative processes, the PSD occupancy in the steady state $B^*/P$ shows a sigmoidal dependence on the concentration of mobile receptors $U/A_{spine}$, which is maintained for different PSD sizes, but more pronounced for large PSDs. Dots: long term average of stochastic simulations. The standard error of the mean is on the order of the dot size (10 trials). Example traces of $B$ for different mobile receptor concentrations and PSD sizes are found in Fig 7D. Dashed lines: mean-field model. **(B)** In the case of cooperative receptor binding, the CV steeply changes with the size and occupancy of the PSD (colored lines) in contrast to the case in which binding rates are constant (black dotted line). **(C)** Shown is the grid of binding sites for different mobile receptor concentrations. Each pixel depicts a single binding site. The pixel color represents the average dwell time of a receptor in the bound state at that specific binding site (1h time course). As we do not explicitly account for diffusion on the grid, positions are kept unitless. The average dwell time $\langle \tau^{PSD}_{dwell} \rangle$ of receptors is not uniformly distributed across the PSD nanodomain. Despite the small size of the nanodomain, as the concentration of AMPARs increases, the receptor dwell times can range from seconds up to tens of minutes at the cluster core (left to right). **(D)** The different dynamics of receptors in the cooperative model are visible in FRAP simulations. Top, left and right: As the PSD size $P$ increases, the recovery of receptors is slowed down. The model matches experimental data of GluA1 recovery in hippocampal cultures [49]. Bottom, left: As the concentration of mobile receptors $U/A_{spine}$ increases, recovery is slowed

down considerably in the cooperative model. At a concentration $U/A_{spine} = 33.4\#/\mu m^2$ the model results match well with experimental data from rat visual cortex [54]. This effect is absent in the non-cooperative model (Bottom, right). Shown are the mean values over 100 trials of the stochastic simulation. The standard error of the mean is in the order of the line thickness.

PSD (Fig 5A). Some effects of cooperative receptor binding should be readily visible in experiments.

1. Cooperativity results in a sigmoidal dependence between PSD occupancy and the concentration of mobile receptors in the spine membrane (Figs 5A and 6A).

2. Whether receptor binding is cooperative could be inferred from the statistics of receptor fluctuations. In the stochastic clustering model (see Methods), the coefficient of variation (CV) of the number of bound receptors $B$ lies within the range of that of EPSC peak amplitudes (Fig 6B, [55]: $CV \approx 20\%–70\%$ with an average of $\approx 39\%$). However, if binding is cooperative, the CV depends on the PSD size and steeply on the occupancy (Fig 6B, colored lines), whereas in the non-cooperative case the CV follows that of a binomially distributed random variable and is independent of the PSD size (black, dotted line). Therefore, steep changes in the CV upon changes in the occupancy of the PSD or changes in the concentration of mobile receptors respectively could indicate cooperativity. It is however difficult to relate fluctuations in e.g. the EPSC amplitude to specific processes such as presynaptic neurotransmitter release or receptor properties.

3. Receptors are not uniformly distributed on nanodomains formed by scaffolding proteins but do form smaller cluster on these (Fig 7F). Such receptor clustering has been found in experiments [39]. We argue that receptor clustering could be the result of cooperative receptor binding.

4. Rigid receptor cluster formation due to cooperative binding results in a non-uniform distribution of receptor dwell times $\tau_{dwell}^{PSD}$ within PSD nanodomains (Fig 6C), where the average dwell time increases with the PSD occupancy ($k_{BU}^{coop} \sim 1/B$; Fig 3B, bottom). Indeed, the time that receptors spend within nanodomains ranges from seconds to a few minutes [39]. Besides, fluorescence recovery after photo-bleaching (FRAP) studies found pools of highly immobile receptors at synapses, which might account for receptors at large cluster cores [49] while receptors within a small cluster or at the border of clusters may account for the very transient immobilization found in single receptor tracking studies [24].

5. Model simulations of FRAP show that, in the cooperative model, the AMPAR recovery is shaped considerably by the concentration of mobile receptors $U/A_{spine}$ compared to the non-cooperative model (Fig 6D, bottom). As before, we assumed a linear relationship between spine area $A_{spine}$ and PSD size $P$ (Eq (3)). Mobile receptor concentrations are varied by a change in the receptor influx $k_{in}$. At low receptor concentrations $U/A_{spine} = 11.2\#/\mu m^2$, AMPAR recovery and its dependence on the spine size and PSD size $P$ are almost indistinguishable between the two models (Fig 6D, top). Fast recovery found in hippocampal cultures [49] is matched at PSD size $P = 81$ and mobile receptor concentration $U/A_{spine} = 11.2\#/\mu m^2$. Slow recovery found in rat visual cortex [54] is matched by the cooperative model at PSD size $P = 144$ and a concentration of $U/A_{spine} = 33.4\#/\mu m^2$ (Fig 6D, bottom left), which is consistent with larger synaptic weights in the visual cortex [56]. However, these results should be interpreted with care when comparing to experimental data as similar results could be achieved by changes in the diffusion coefficient or the receptor

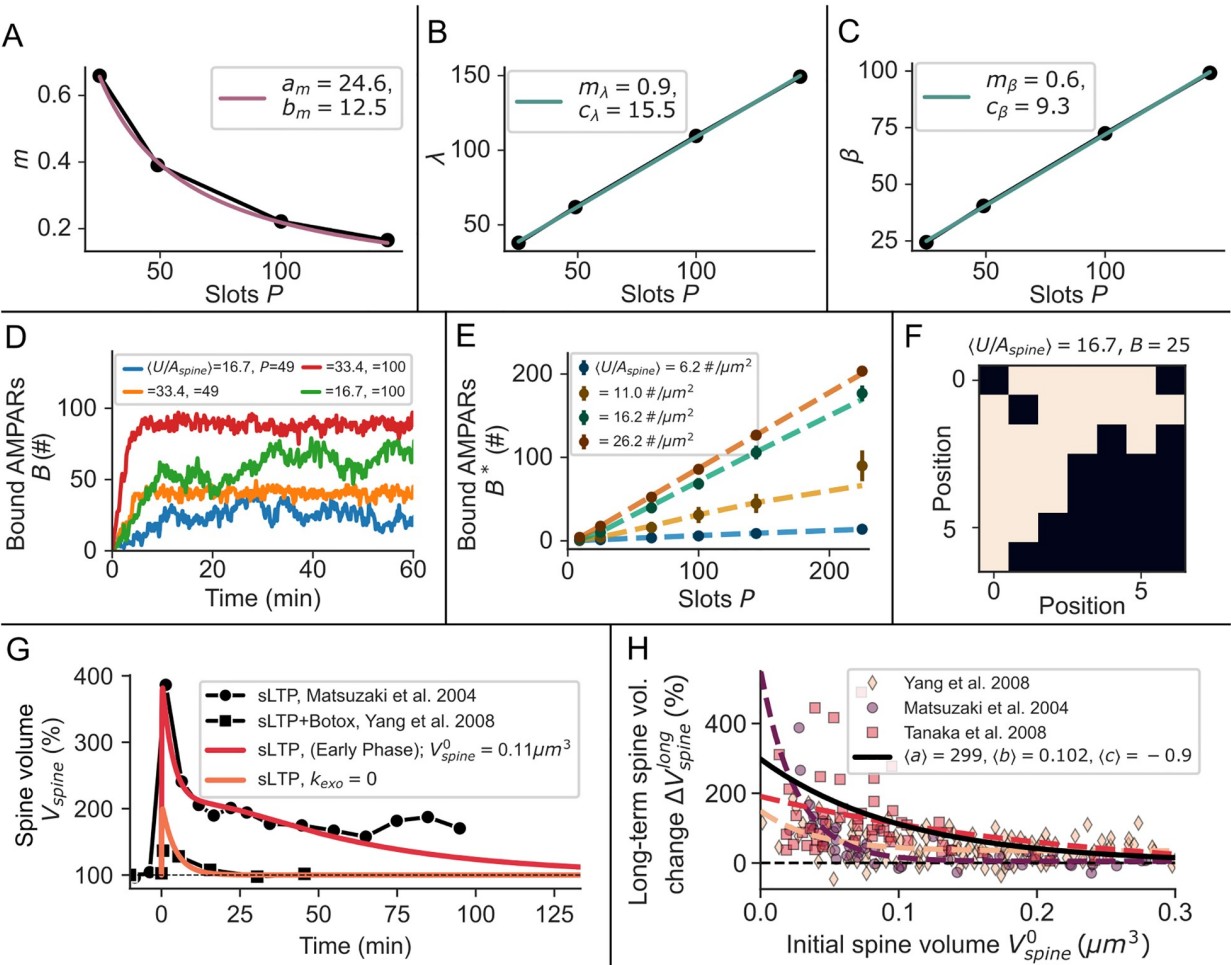

**Fig 7. Functional relations in the stochastic grid model of cooperative binding and spine size dependence of sLTP. (A)** The slope $m(P)$ (Eq (22)) of the relation between the mean cooperative binding rate $\langle \hat{k}_{UB}^{coop} \rangle$ and the number of bound receptors $B$ depends on $P$, as on small grids the mean number of nearest neighbours increases faster than on large grids. Black dots: values derived from fitting Eq (19) to stochastic simulation data for different values of $P$. Colored line: fit of Eq (22) to data points (black dots). **(B,C)** The mean cooperative unbinding rate $\langle \hat{k}_{BU}^{coop} \rangle$ depends on $B$ and $P$ (Eq 18) with functions $\lambda(P)$ and $\beta(P)$ depicted in (B) and (C). Black dots: same as in (A) but for fitting Eq (18). Colored lines: same as in (A) but for fitting Eqs (21) and (20). **(D)** Example traces of the number of bound AMPARs $B$ for different average mobile receptor concentrations $\langle U/A_{spine} \rangle$ and PSD sizes $P$. **(E)** The steady state value of $B$ depends linear on the PSD size $P$. The mean-field model (dotted lines) matches the stochastic model (dots). Bars indicate standard deviations. **(F)** Example of a cluster realization in the stochastic model where white squares denote occupied slots. **(G)** Spine volume increase during sLTP in the model (colored lines) compared to experimental data taken from [19, 20]. **(H)** The relative increase in spine volume depends on the initial spine size. Dashed lines: Eq (11) has been fitted to data from [19, 20, 34] (marker). Black line: Parameters $a$, $b$, $c$ (see Eq (11)) averaged across the individual estimates (dashed lines).

unbinding rate. Still, this shows that cooperativity affects recovery and therefore could be inferred from FRAP experiments.

6. We assumed an ongoing, rapid exchange of bound receptors, which has been observed experimentally [24]. However, some receptors may be much more stable [49]. We predict that such a stable receptor pool is not necessary for E-LTP expression. This prediction could be tested experimentally by blocking exocytosis as well as diffusion into the spine to interrupt the mobile receptor influx. If receptors are not stabilized otherwise, such depletion of the mobile receptor pool should result in the total decay of synaptic transmission within minutes.

7. We find that there exist three alternative explanations for the dependence of E-LTP on exocytosis: E-LTP is maintained by 1) stabilization of receptors due to an exocytic factor [53], 2) an exocytic factor that raises the probability of receptor binding or 3) a rapid increase in the number of slots after LTP induction [57, 58]. Most apparent, neither the generation of additional binding sites nor the exocytosis of factors that stabilize AMPARs require an increase in the concentration of the spine's mobile receptor pool to maintain E-LTP, whereas in our model the mobile pool concentration needs to grow by ≈25% (Fig 3D). We predict that this increase could be mediated by a sLTP dependent change in receptor exocytosis at the spine (Fig 3C and 3D).

## Discussion

The question of how E-LTP is maintained has not been answered in previous studies. Also, the role of receptor exocytosis is still ambiguous. In this study, we focused on two major experimental observations: First, E-LTP can be long-lasting with decay times on the order of hours [5, 59]. Second, there exists a strong dependency between LTP and exocytosis, although the exocytosis of AMPARs is slow compared to the rapid exchange of receptors via lateral diffusion [11, 12, 15, 16, 60]. Our model indicates that the experimentally observed increase in the rate of AMPAR exocytosis events and their binding to the PSD scaffold are not sufficient to maintain E-LTP (Figs 2 and 4 and S1 Fig). Furthermore, our results indicate that sLTP accompanied by increased AMPAR trafficking from REs during E-LTP [29, 38, 61] and cooperative receptor binding [4, 45] individually also can not account for the maintenance of E-LTP. However, we demonstrated and predict that cooperative receptor binding complemented by increased trafficking from REs is sufficient to sustain E-LTP. Thus, our study provides several insights about the functional roles of diverse mechanisms during LTP and yields predictions highlighted above.

### Bound AMPARs as the determinant factor of synaptic strength

We assumed that the EPSP amplitude measured in LTP experiments is determined by the number of bound AMPA receptors. This assumption is supported by several experimental studies demonstrating that LTP is mostly expressed at the postsynaptic site and that excitatory synaptic transmission depends on the number of AMPARs [15, 62–65]. However, the relationship between receptor number and EPSP amplitude may be nonlinear. For example, the alignment of receptors with the presynaptic release site is an important factor that influences the postsynaptic response [39, 66, 67]. However, whether or to what degree enhanced alignment might occur during E-LTP is unknown. Other important factors that shapes the postsynaptic response are the presynaptic release probability and the amount of glutamate that is released. Depending on the experimental setup and the conditions of LTP induction, long-lasting presynaptic changes sometimes contribute to LTP [68, 69]. However, several studies have shown that LTP is often independent of presynaptic changes and thus mainly driven postsynaptically [15, 62, 69]. Therefore, we target the question of whether changes in AMPAR trafficking alone could be sufficient for E-LTP expression. In addition, several types of AMPARs have been identified that may differently contribute to LTP and synaptic transmission [49, 70–76]. The overall conductance could be adapted by the insertion of different AMPAR types, each having a different conductivity [77]. The calcium-permeable GluA1 homomer (CP-AMPAR) for example has a conductance that is approximately four times higher than that of the GluA1/2 heteromer [77]. However, CP-AMPARs are not required for E-LTP expression [78] such that

we limited our model to one type of receptor. Also, AMPARs already present at the PSD before induction might change their conductivity [79]. Stargazin, for example, has been shown to enhance the glutamate affinity and conductance of AMPARs [80, 81]. By contrast, the results from [15] indicate that the fast increase after LTP-induction is associated with the binding of additional AMPARs and possibly not with changes of receptor conductivity. Thus, in this study, we do not consider conductance changes of individual AMPARs. However, different levels of conductivity can be integrated into our model, for instance by weighting the contribution of an AMPAR to the synaptic strength, and by this, the influence of temporal variations in the conductivity by the possible exchange of AMPAR types on LTP could be investigated.

## AMPAR mobility

We assumed that AMPA receptors are transiently immobilized to become functional at the PSD and otherwise rapidly diffuse between spine and dendrite. This assumption is supported by several experimental studies [3, 4, 82]. However, the experimental literature is not clear regarding the dwell time of immobilized receptors. Studies blocking exocytosis [15, 16, 60] show that synaptic potentiation is very transient, which implies that newly bound AMPARs may unbind from binding sites within seconds or a few minutes and diffuse out of the PSD. Also, single-receptor-tracking studies revealed AMPAR dwell times in immobilized or confined states on the order of seconds [24, 39, 51, 83]. However, the estimates from these studies are biased by the fact that receptors are tracked for only 60–200 *s*. In contrast, studies investigating the fluorescence recovery after photo-bleaching (FRAP) suggest that immobile receptors are trapped for 30 minutes and longer [35, 49, 50]. However, it is not clear whether the pool of immobilized receptors measured in FRAP experiments is affected by LTP-induction. While one study found that the number of immobile GluA1, but not GluA2 increases after 2-photon glutamate uncaging [49], another study did not find a significant increase [12]. Furthermore, single-receptor-tracking as well as FRAP experiments use probes, which could physically interact with the system and, thus, impede the evaluation of receptor dwell times. While the role of Stargazin and MAGUKs such as PSD95 in receptor accumulation has been extensively studied [39, 83–86], the entirety of mechanisms that drive receptor immobilization during synaptic plasticity is still unclear. In addition to changes in phosphorylation states of PSD proteins, different states of immobilization could be mediated by clustering, as shown in this study, but also by adaptation in steric hindrance/obstacle density via other proteins [87] or some, possibly exocytic, factors that alter the affinity between receptors and binding sites. To match the experimental observation that E-LTP rapidly decays in the absence of exocytosis, we set the time scale for the change in receptor binding $k_{UB}$ to be small, which matches well with the decay of CaMKII activity after LTP induction. However, one should be careful when interpreting these results. Whereas this could be seen as an indicator for rapid dephosphorylation of synaptic proteins such as TARPs, the phosphorylated state may as well outlast CaMKII activity. The rapid decay in EPSP amplitude could also be attributed to the need for an exocytic factor or other processes that may require exocytosis dependent structural plasticity such as translocation of CaMKII or PSD reorganization. More experimental work is required.

## PSD growth during LTP

We assumed no change in the number of binding sites during E-LTP, which is consistent with experimental observations showing that an increase in PSD size is protein synthesis dependent and only occurs about 50 minutes after LTP induction [27]. Also the number of PSD95 increases very gradually over the time course of 3 hours after LTP induction and only if the spine enlargement is stable [28]. However, it has been hypothesized that new binding sites

may become available due to the unbinding of the protein SynGAP from PSD95 upon phosphorylation [4, 57, 88, 89]. Thereby, the generation of new slots may also contribute to E-LTP maintenance. In addition, the CaMKII-NMDAR complex might mediate the composition of new binding sites [90]. However, more data is required to conclusively quantify the influence of PSD changes on the maintenance of E-LTP.

## Correlation between RE trafficking, structural and functional LTP

How sLTP, RE trafficking, and functional LTP are interlinked is still ambiguous. Several experimental studies identified correlations between spine volume and synaptic transmission [19, 20, 33]. For instance, [19] have shown that AMPAR-mediated currents and spine volume increase during LTP are correlated in the very early phase of LTP (1–5 $min$) as well as at later stages ($>30$ $min$). These results indicate that the enlargement of the spine head might be an essential condition for the expression of LTP. In addition, exocytosis from AMPAR-containing REs is required for spine size maintenance [20, 29], and these are mobilized into the spine upon LTP-induction [29–31, 38]. Also, exocytosis is required for sustained E-LTP [15]. In this study, we, therefore, assumed that the spine volume is the determinant factor for the number of AMPARs exocytosed from REs during an exocytosis event. However, whether and how exocytosis scales with the spine size is unclear and additional experimental data is needed. Also, it is unclear whether exocytosis of AMPARs or other cargo is the major factor shaping receptor accumulation [53] and it has been suggested that REs are required for the regulation of synaptic PSD-95 levels rather than AMPAR recycling [91]. Besides, there have been conflicting results regarding the impact of REs during the initial phase of sLTP [29, 91].

## Location of exocytosis

LTP not only influences exocytosis at the spine but also at the dendrite [12, 18]. It is however unclear whether and how receptors that are exocytosed outside the spine contribute to E-LTP expression. Theoretical studies have shown that the locus of exocytosis close to the PSD is important for synaptic capturing of AMPARs [36, 92]. A sustained increase in the exocytosis rate of GluA2 and GluA3 after LTP induction has been observed experimentally at the dendritic shaft [18]. Such a sustained local increase in dendritic receptor concentration close to a spine might increase the receptor flux ($k_{in}$) sufficiently to supplant or support RE mediated receptor recycling at spines and thereby contribute to E-LTP expression.

As our model does not explicitly account for spatial properties, it is not well suited to investigate such dependencies. However, changing the location of exocytosis should alter the effective contribution of exocytosis to the spine's mobile pool, meaning that the fraction of the exocytic input that reaches the target site changes. If receptor exocytosis occurs close to the dendritic shaft, we expect a smaller proportion of these receptors to dwell in the spine and reach the PSD compared to exocytosis occurring at the spine. Also, as receptor binding is diffusion limited, exocytosis occurring close to the PSD would therefore increase the probability of those receptors to get bound. This effective change in the contribution of exocytosis to the receptor content at the spine could simply be accounted for by changing, for example, the exocytosis event size $S_{exo}$ or by introducing an additional parameter that accounts for the limited availability of the input to the target site (by changing $S_{exo} k_{exo}$ into $S_{exo} k_{exo} b_{exo}$, where $b_{exo}$ describes the availability of the exocytic input to the mobile receptor pool). This dependence between exocytosis event size and bound receptor number is shown in Fig 5A. Due to it's sigmoidal nature, a small shift in $S_{exo}$ would already significantly lower the number of bound receptors in our model. Therefore, in agreement with other theoretical studies [36, 92], our model supports the importance of exocytosis occurring close to the spine.

### Exocytic factor hypothesis

The dependence of LTP on exocytosis may rely on specific exocytic factors that the model does not account for. Such factors could be specific transmembrane proteins that stabilize receptors at the PSD or that increase the binding probability or even contribute to the generation of additional binding sites. Such a contribution of an exocytic factor has been hypothesized in recent reviews [4, 53]. An intriguing hypothesis is that the affinity of PSD slots for AMPARs depends on their subunit composition and/or posttranslational modifications [93]. Exocytosis of such AMPARs with different binding properties may be required for E-LTP. Such a model could also explain E-LTP maintenance and its rapid decay in the absence of exocytosis. We show this in an exemplary model in the supplementary material (S1 Appendix).

### Signaling pathway models

Complex signaling pathway circuits underlie many cell processes such as synaptic plasticity. In this study, we did not model such pathways explicitly, but instead consider only their effect on AMPAR trafficking, as the time course of parameter values is determined by signalling dynamics. By this, our model becomes concise and mathematically feasible, but it does not capture complex signalling dynamics such as oscillations or switching between bistable states [94, 95]. Please note that, if affecting AMPAR trafficking, such complex signalling dynamics should be contained in the experimentally observed time course of AMPAR exocytosis, mobility or structural plasticity being considered in this study. However, it should also be noted that the available experimental data delivers an incomplete picture such that the integration of signaling pathways models [94, 95] into our model of AMPAR trafficking could be an interesting next step to further deepen our understanding on how spine structure, enzymes and ion channels interact during LTP.

### Conclusion

In line with several other studies [30, 31, 36, 39, 42, 43, 45], our results support the importance of cooperative binding, spine morphology, and RE trafficking for LTP. Beyond this, our model predicts that cooperative binding and spine morphology complement each other to sustain E-LTP such that other phases or processes of (late-phase) LTP can set in. As discussed above, the processes involved in the control of AMPAR mobility and trapping, in particular concerning morphological spine changes and exocytosis, remain to be fully elucidated. The model derived in this study provides a foundation for further investigations of these processes and could be applied to other types of synaptic plasticity such as LTD and synaptic scaling or the role of different AMPAR types. Proper functioning of various types of synaptic plasticity is important for learning and memory and dysregulation of processes involved in synaptic plasticity such as AMPAR trafficking have been implicated in different diseases such as Alzheimer's and epilepsy [96]. To understand how deficits in receptor trafficking can evolve into neurodegenerative diseases we need to obtain a profound understanding of how different trafficking mechanisms can influence synaptic transmission.

## Materials and methods

### Parameter derivation

All model parameters are listed in Tables 1 and 2. In the following, we will motivate the different parameter choices. Parameter values under basal conditions, i.e. before LTP induction, will be denoted by a superscript index 0.

The *number of AMPARs* in the PSD has been estimated at around 15–30 receptors [39, 48, 52, 97] (the maximum number could be up to 140 receptors [33, 48]). The exact fraction of mobile receptors in the spine is unclear. While some studies report around 50% mobile receptors at spines [23, 35, 83, 98] more recent studies show that $\approx$65–90% of the AMPARs are confined in PSD nanodomains [39, 99]. The question remains how many of these confined AMPARs are also contributing to the excitatory transmission or whether they functionally belong to the mobile pool $U$ in our model. Considering the above here we assume a total number of 30 AMPARs in the steady state with 2/3 bound receptors ($B^* = 20$) and 1/3 mobile receptors ($U^* = 10$); $^*$ marks the steady state.

The *exocytosis event rate* within spines has been measured by Patterson et al. [12] who found $\approx$0.11 exocytosis events per minute under basal conditions ($k_{exo}^0$). It has been estimated that about 51±36 GluA1 subunits are released into the membrane upon an exocytosis event at the dendritic shaft [47]. However, exocytosis events at spines are on average smaller than at the dendritic shaft [12]. Another study by [18] only estimated 10–15 subunits per event. AMPARs are tetrameric, containing either one or two types of subunits. Based on the above findings we estimated the exocytosis event at spines to have a size of about $S_{exo}^0 = 13$ receptors.

The *diffusion-dependent parameters* $k_{out}$ and $k_{in}$ are estimated from FRAP and receptor tracking experiments, assuming that the contribution of exocytosis to the fluorescence recovery is small [11, 12, 49]. The recovery half time in these FRAP experiments ranges from tens of seconds up to 100 seconds depending on the spine and AMPAR subunit type [12, 35, 49, 50, 54]. We, therefore, estimate that diffusing AMPARs enter the spine's membrane from the dendrite at a rate of $k_{in}' \approx 0.02 s^{-1}$. Given that receptor exchange occurs mainly due to diffusion [12, 49], in- and efflux need to be properly balanced in accordance with the steady state value $U^*$. Therefore, we set $k_{in} = k_{in}' \cdot U^*$. Single receptor tracking and theoretical studies reveal receptor dwell times in confined states at spines that are also on the order of tens of seconds [22, 36, 39, 51]. Therefore $k_{out} = k_{in}' A_{spine}^0$ is a convenient choice. Alternatively, we will sometimes use the term receptor dwell time $\tau_{dwell} = A_{spine}/k_{out}$.

The *unbinding rate* of AMPARs $k_{BU}$ is estimated based on the dwell time of receptors being in an immobile state at the PSD. Given [24], we set $k_{BU} = 1/10\ s^{-1}$. We assume that functional AMPARs need to be in an immobilized state, considering that precise alignment of AMPARs with the presynaptic active zone is important for excitatory synaptic transmission [100].

The number of binding sites or *slots P* contained in a PSD is not clear. However, the receptor number in spines is much smaller than the number of most scaffolding proteins being in the PSD [52]. Also, the EPSP amplitude increases by about 50–200% during LTP-induction, which is consistent with the increase in GluA1 fluorescence [27]. Considering that $\approx$60–70% of the receptors are located within PSD Nanodomains and from the assumption that the PSD does not change during E-LTP follows that the PSD is not saturated before induction. Therefore, to account for the 200% increase in AMPARs, we set $P = 3.5 \cdot B^*$.

For the average *spine volume* under basal conditions we assume $V_{spine}^0 = 0.08 \mu m^3$ [19, 37]. We calculate the surface area $A_{spine}^0$ from the volume considering spherical spine heads.

Given that we set the number of AMPARs under basal conditions, the *endocytosis rate* $k_{endo}$ and the *binding rate* $k_{UB}^0$ (before LTP induction) can be calculated analytically from the fixed point equation of the basic model:

$$k_{endo} = \frac{A_{spine}^0 \left(k_{exo}^0 S_{exo}^0 + k_{in}\right) - k_{out} U^*}{U^*}, \tag{4}$$

$$k_{UB}^0 = \frac{A_{spine}^0 \, k_{BU} \, B^*}{(P - B^*) \, U^*}. \tag{5}$$

**Parameter changes during LTP-induction.** LTP-induction triggers a change in the exocytosis event rate $k_{exo}$ (*Events/s*) and the binding rate $k_{UB}$ [12, 15, 21, 84] (see Introduction and Fig 1). The change in both parameters $k_x$, $x \in \{exo, UB\}$ during LTP-induction is modeled by the following functions:

$$k_x(t) = k_x^0 \cdot f(t, C_x = 1), \tag{6}$$

with

$$f(t) = \left( C_x + A_x \frac{e^{-t/\tau_x^2} - e^{-t/\tau_x^1}}{A_{norm}} \right) \tag{7}$$

where

$$A_{norm} = \left( \frac{\tau_x^2}{\tau_x^1} \right)^{\frac{\tau_x^1}{\tau_x^1 - \tau_x^2}} - \left( \frac{\tau_x^2}{\tau_x^1} \right)^{\frac{\tau_x^2}{\tau_x^1 - \tau_x^2}} \tag{8}$$

is a normalizing factor, such that $max((exp(-t/\tau_x^2) - exp(-t/\tau_x^1))/A_{norm}) = 1$. $A_x$ is the factor by which the parameter changes during LTP-induction. $\tau_x^2$ and $\tau_x^1$ are time constants. Upon LTP-induction the exocytosis event rate increases 5-fold for the duration of the stimulus ($\approx 1$ *min.*), as shown in 2-photon-glutamate-uncaging experiments [12]. We therefore assume a rapid increase in the exocytosis event rate $k_{exo}$ with time constants $\tau_{exo}^1 = 25s$ and $\tau_{exo}^2 = 60s$ during E-LTP-induction. Based on [12, 13], we set $A_{exo} = 5$.

During LTP induction, AMPARs are transiently trapped at the synapse [15], indicating that the binding rate $k_{UB}$ is transiently increased. Therefore, we set $\tau_{UB}^1 = 5s$ and $\tau_{UB}^2 = 60s$. Consistent with this, CaMKII activity rapidly increases but decreases within 2 *min* upon 2-photon glutamate uncaging [21, 101].

There exists no clear experimental data from which the change in amplitude of $k_{UB}$ during LTP induction could be inferred. Thus, we assume that the increase is large enough to nearly saturate the available slots, which is the case for $A_{UB} = 30$ in the non-cooperative model and $A_{UB} = 5$ in the cooperative model.

## Computer simulation and random sampling of the parameter space

For each model version, we used the classical Runge-Kutta method to solve the corresponding set of differential equations.

In addition to the parameter estimates from above, we randomly sampled parameters (Fig 4 and S1 Fig) to evaluate within which parameter regime the model can be matched to the experimental data of E-LTP with and without exocytosis. In this method, parameters of the model are chosen randomly within a predefined regime (Table 3). The outcome of a simulation is compared to the reference experimental data using the following cost function:

$$\Delta x = \frac{1}{T} \left( \int_0^T |B_{ref}^{E-LTP} - B_{sim}^{E-LTP}| dt + \int_0^T |B_{ref}^{k_{exo}=0} - B_{sim}^{k_{exo}=0}| dt \right), \tag{9}$$

where $B_{ref}^{E-LTP}$ (with exocytosis), $B_{ref}^{k_{exo}=0}$ (without exocytosis) are the reference data to which we match the model and $T$ is the duration of measurements. The cost function can be interpreted

as the time-averaged difference in the number of bound receptors between modeling and experimental data. We considered 2000 trials (different sets of parameter values) and evaluated the best 0.5%, according to the cost function.

We used as reference the experimental data from [10] and [15] (E-LTP, LTP with blocked exocytosis; Fig 1A), which is given in percentage of the baseline EPSPs. We fitted the EPSP data by the following function:

$$
\begin{aligned}
EPSP^x(t) \quad &= f(t, C^x_{short}, A^x_{short}, \tau^{x,2}_{short}, \tau^{x,1}_{short}) \\
&+ f(t, C^x_{long}, A^x_{long}, \tau^{x,2}_{long}, \tau^{x,1}_{long}),
\end{aligned}
\tag{10}
$$

where $f(\cdot)$ is defined by Eq (7) and $x \in \{E - LTP, k_{exo} = 0\}$. The function $EPSP^x(t)$ accounts for a short initial phase as well as for a subsequent long lasting phase. We transferred this data from % of baseline EPSPs into units of receptor numbers by assuming the same baseline number of receptors for the reference as we did for the model ($B^x_{ref} = \frac{EPSP^x(t)}{100\%} \cdot B^*$).

## Model of sLTP

In the following we describe our model of sLTP and its influence on AMPAR exocytosis from recycling endosomes at the spine [29, 30]. While much data on spine size dynamics is available, there exists little data on the recycling endosome dynamics in comparison. However, experimental data indicates that both are tightly coupled [29]. Therefore, we first formulate a model of the spine volume during sLTP and then the dependence of exocytosis ($S_{exo}$) on the spine volume.

Similar to functional LTP, sLTP can be divided into different phases [19]. Directly after the induction of sLTP the spine volume rapidly increases and subsequently decays ($V^{short}_{spine}$) but settles at a level above the original baseline ($V^{long}_{spine}$). sLTP and functional LTP are closely correlated and in the absence of protein synthesis the late phase of sLTP ($V^{long}_{spine}$) decays with a time constant similar to E-LTP [20, 27]. In addition, sLTP depends on exocytosis and rapidly decays after application of Botox [20, 29] (Fig 1C). Furthermore, several experiments suggest that the relative increase in spine volume triggered by LTP-induction $\Delta V^{long}_{spine}$ depends on the initial spine volume $V^0_{spine}$ [19, 20, 34]). The experimental data indicates that sLTP expression is strongest for small to intermediate sized spines but decays as $V^0_{spine}$ increases such that there is no long-term change in volume for large spines [19]. We approximate this relationship by an exponential function:

$$
\Delta V^{long}_{spine} = \frac{a \cdot e^{V^0_{spine}/b} + c}{100\%},
\tag{11}
$$

where $a + c$ determines the maximum relative change in spine volume $\Delta V^{long}_{spine}$ and $c$ accounts for the possibility of large spines experiencing structural depression or potentiation. $b$ describes the decay of $\Delta V^{long}_{spine}$ with the initial spine size $V^0_{spine}$. We estimated these parameter values for three different studies [19, 20, 34] by fitting Eq (11) to the experimental data. For the final model, we averaged parameters across the different estimates (values are given in the legend of Fig 7H).

Based on the findings discussed above, we formalized a model of the spine volume change:

$$
\begin{aligned}
V_{spine}(t) \quad &= V^0_{spine} \cdot [f(t, 1, 1 + 1.75 \cdot \Delta V^{long}_{spine}, \tau^2_{short}, \tau^1_{short}) \\
&+ f(t, 0, \Delta V^{long}_{spine}, \tau^2_{long}, \tau^1_{long})],
\end{aligned}
\tag{12}
$$

where $f(\cdot)$ is defined by Eq (7). The first term on the right hand side describes the initial, transient phase of structural plasticity $V_{spine}^{short}$ that decays within 10–20 $min$ ($\tau_{short}^1 = 5s$, $\tau_{short}^2 = 250s$). The amplitude depends on $\Delta V_{spine}^{long}$, but it has a lower limit in the absence of a late phase $(1 + 1.75 \cdot \Delta V_{spine}^{long})$ [19]. The second term on the right hand side describes the long lasting second phase $V_{spine}^{long}$ ($\tau_{long}^1 = 500s$). When L-LTP is triggered, $\tau_{long}^2$ is very large, i.e. the spine volume does not decay over the time course of hours. In the absence of L-LTP, we set $\tau_{long}^2 = 50min$, which is the decay time constant of E-LTP. If exocytosis is blocked, $\Delta V_{spine}^{long}$ is set to zero. Thereby, our model closely matches the time course of sLTP for different spine sizes and conditions (Fig 7G). To account for endosomal trafficking during sLTP we describe the dynamics of the exocytosis event size $S_{exo}$ at the spine by

$$\frac{dS_{exo}}{dt} = k_{in}^{RE} - k_{out}^{RE}\,\frac{S_{exo}}{V_{spine}(t)}, \tag{13}$$

where we set $k_{in}^{RE} = 0.1\#s^{-1}$. From the fixed point equation of (13) we derive $k_{out}^{RE} = k_{in}^{RE}\,V_{spine}^0/S_{exo}^* = 0.000615\,\mu m^3/s$ (where $S_{exo}^* = S_{exo}^0 = 13$, see Methods section on parameter derivation).

## Cooperative binding model

Thanks to super-resolution microscopy techniques like STORM and STED, in recent years, several studies revealed that synaptic proteins in the PSD are not homogeneously distributed but often clustered in so-called nanodomains [26, 39, 40, 102]. Several theoretical studies identified that clusters can emerge due to cooperative binding of proteins [41–45, 103]. PSD scaffolding proteins such as PSD95 form nanodomains on which AMPARs are not uniformly distributed but form smaller clusters themselves [39], indicating cooperative binding of receptors.

To extend the basic model by cooperative binding of AMPARs in the PSD we utilize a recent theoretical model capturing the essence of cooperativity given a few general assumptions [45]. In our adaptation of this model the scaffolding proteins or slots are spatially organized on a square grid (Fig 3A) where each mobile receptor $U$ binds to a free slot with a certain rate, dependent on the number of occupied nearest-neighbour slots (Eq (14)). Note that one could use different types of grids, e.g. a hexagonal grid, to account for a different maximum number of nearest neighbours. Similarly, the rate of unbinding decreases with more bound receptors $B$ being in the surrounding (Eq (15)). Note that in this model the grid size $(N \times N)$ is analog to $P$ in the basic model (Eqs (1) and (2)).

$$\hat{k}_{UB}^{coop} = k_{UB}(\alpha\chi + 1) \tag{14}$$

$$\hat{k}_{BU}^{coop} = k_{BU}(1 - \chi). \tag{15}$$

Here, $\chi$ is the fraction of occupied nearest neighbours on the grid (i.e. $n/8$ with $n = 0, 1, \ldots,$ 8) and $\alpha$ is a measure for the cooperativity, where a big value coincides with a stronger influence of cooperativity. Taking into account the dependence on the mobile receptor concentration the total binding flux is $\hat{k}_{UB}^{coop}\,\frac{U}{A_{spine}}$. As apparent from Eqs (14) and (15), the cooperative binding and unbinding rate $\hat{k}_{UB}^{coop}$, $\hat{k}_{BU}^{coop}$ change depending on the number of bound receptors on the grid and depending on the size of the grid $P$. We did a stochastic simulation of this model (as in [45]) with the binding and unbinding probabilities being $p_{UB}^{coop} = \hat{k}_{UB}^{coop}\,\frac{U}{A_{spine}} \cdot \Delta t$ and

$p_{BU} = \hat{k}_{BU}^{coop} \cdot \Delta t$. The integration time step was set between $\Delta t = 0.1\ s$ and $\Delta t = 0.5\ s$. In addition, we describe the mobile receptor pool $U$ by stochastic simulation of the corresponding differential equation Eq (1). To control the number of mobile receptors, and thereby, also the number of bound receptors on the grid, we vary the influx $k_{in}$ between 0.1–1.2#/s. The binding parameter were set to $k_{UB} = 0.0005\mu m^2(\#s)^{-1}$, $\alpha = 16$ and $k_{BU} = 0.1\ s^{-1}$.

To integrate the key principles of the cooperative model into the basic model of AMPAR-dynamics (Eqs (1) and (2)), we evaluated several iterations of the cooperative model for different numbers of available receptors $U$ and slots $P$ (grid size). From these, we obtained the mean binding $\langle \hat{k}_{UB}^{coop} \rangle := \langle \hat{k}_{UB}^{coop} \rangle_{B,P}$ and unbinding rate $\langle \hat{k}_{BU}^{coop} \rangle := \langle \hat{k}_{BU}^{coop} \rangle_{B,P}$ per mobile/bound receptor averaged across the grid and across many realizations having a number of slots $P$ and a number of bound receptors $B$ (Eqs (16) and (17)). Thereby, we acquire the dependence of $\langle \hat{k}_{UB}^{coop} \rangle$ and $\langle \hat{k}_{BU}^{coop} \rangle$ on $B$ and $P$ (Fig 3B):

$$\langle \hat{k}_{UB}^{coop} \rangle = \frac{1}{n_P^0} \sum_{\substack{\Omega^0 \in N_P^0 : \\ |\Omega^0| = P - B}} \frac{1}{P - B} \sum_{slot \in \Omega^0} \hat{k}_{UB}^{coop}(slot)\ , \tag{16}$$

$$\langle \hat{k}_{BU}^{coop} \rangle = \frac{1}{n_P^1} \sum_{\substack{\Omega^1 \in N_P^1 : \\ |\Omega^1| = B}} \frac{1}{B} \sum_{slot \in \Omega^1} \hat{k}_{BU}^{coop}(slot)\ , \tag{17}$$

where $\Omega^0$ and $\Omega^1$ denote the sets of empty and occupied grid elements for one realization respectively. $N_P^0$ and $N_P^1$ contain the sets of empty and occupied grid elements for all realizations with a total number of grid elements $P$. $n_P^0$ and $n_P^1$ denote the number of realizations for which the conditions $|\Omega^0| = P - B$ and $|\Omega^1| = B$ hold. Note that the choices for $k_{UB}$ and $k_{BU}$ do not influence the dependence of $\langle \hat{k}_{UB}^{coop} \rangle$ or $\langle \hat{k}_{BU}^{coop} \rangle$ on $P$ or $B$.

We then formalized the dependency of binding and unbinding rate on $P$ and $B$ by fitting $\langle \hat{k}_{UB}^{coop} \rangle$ and $\langle \hat{k}_{BU}^{coop} \rangle$ by the following functions (Fig 3B dashed lines):

$$\frac{k_{BU}^{coop}}{k_{BU}} = \frac{\lambda(P)}{\beta(P) + B} - 0.5, \tag{18}$$

$$\frac{k_{UB}^{coop}}{k_{UB}} = m(P)B^{0.8} + 1 \tag{19}$$

with functions

$$\beta = m_\beta P + c_\beta, \tag{20}$$

$$\lambda = m_\lambda P + c_\lambda, \tag{21}$$

$$m = a_m/(P + b_m). \tag{22}$$

$\beta(P)$, $\lambda(P)$ and $m(P)$ are shown in Fig 7A–7C. By this, we eliminate the spatial component of the cooperative binding model and obtain a mean-field description that we can integrate into our basic model by substituting $k_{UB}$ and $k_{BU}$ in Eqs (1) and (2) by $k_{UB}^{coop}$ and $k_{BU}^{coop}$ (Fig 3B). The mean-field model closely matches the stochastic spatial clustering model's static and dynamic properties (Figs 6A, 7E and 3A inset).

**Steady state binding rate in the cooperative model.** The dynamics of cooperative binding changes the steady state of the system. Therefore, the binding rate has to be derived from the the adapted fixed point equation resulting to:

$$k_{UB}^0 = \frac{A_{spine}^0 \, B^* \, k_{BU} \left( \frac{c_\lambda + m_\lambda \, P}{B^* + c_\beta + m_\beta \, P} - 0.5 \right)}{(P - B^*) \left( 1 + \frac{a_m \, (B^*)^{0.8}}{b_m + P} \right) U^*}. \tag{23}$$

## Code accessibility

Code and documentation are available on GitHub https://github.com/MoritzB90/AMPAR-Trafficking-Model.

## Supporting information

**S1 Fig. The parameter values of exocytosis for the best 0.5% parameter sets depend on the number of bound AMPARs under basal condition and the dwell time of mobile receptors at the spine.** For the three models of AMPAR-dynamics (first column: basic model; second column: sLTP model; third column: cooperative binding model), the exocytosis event rate under basal condition ($k_{exo}^0$; first row), the maximal amplitude of LTP-induced exocytosis event rate ($k_{exo}^A = k_{exo}^0(1 + A_{exo})$; second row), and the decay time of the exocytosis event rate after LTP-induction ($\tau_{exo}$; third row) are shown. Black lines depict standard deviations. The red areas indicate the biologically plausible regime of the corresponding indicators based on experimental studies [11, 12, 47].
(PDF)

**S1 Appendix. Exocytic factor.**
(PDF)

## Acknowledgments

We would like to thank Silvio Rizzoli for various fruitful discussions and helpful comments.

## Author Contributions

**Conceptualization:** Moritz F. P. Becker, Christian Tetzlaff.

**Data curation:** Moritz F. P. Becker.

**Formal analysis:** Moritz F. P. Becker.

**Funding acquisition:** Christian Tetzlaff.

**Investigation:** Moritz F. P. Becker, Christian Tetzlaff.

**Methodology:** Moritz F. P. Becker, Christian Tetzlaff.

**Project administration:** Christian Tetzlaff.

**Resources:** Christian Tetzlaff.

**Software:** Moritz F. P. Becker.

**Supervision:** Christian Tetzlaff.

**Validation:** Moritz F. P. Becker, Christian Tetzlaff.

**Visualization:** Moritz F. P. Becker.

**Writing – original draft:** Moritz F. P. Becker, Christian Tetzlaff.

**Writing – review & editing:** Moritz F. P. Becker, Christian Tetzlaff.

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
