## [Decision Letter · Decision Letter 0]

2 Dec 2020

Dear Mr. Becker,

Thank you very much for submitting your manuscript "The biophysical basis underlying the maintenance of early phase long-term potentiation" for consideration at PLOS Computational Biology.

As with all papers reviewed by the journal, your manuscript was reviewed by members of the editorial board and by several independent reviewers. In light of the reviews (below this email), we would like to invite the resubmission of a significantly-revised version that takes into account the reviewers' comments.

We cannot make any decision about publication until we have seen the revised manuscript and your response to the reviewers' comments. Your revised manuscript is also likely to be sent to reviewers for further evaluation.

Sincerely,

Joanna Jędrzejewska-Szmek, Ph.D.

Guest Editor

PLOS Computational Biology

Kim Blackwell

Deputy Editor

PLOS Computational Biology

Reviewer's Responses to Questions

**Comments to the Authors:**

Reviewer #1: Becker and Tetzlaff investigated an important phenomenon occurring in neuronal synapses, that is the long-lasting modification of the number of receptors for neurotransmitters due to synaptic plasticity mechanisms. These mechanisms, which are thought to underly learning and memory, are not completely understood from the physico-chemical point of view as experimental data at the molecular level is heavily constrained by technical issues. Simulations and theoretical interpretations are helpful approaches for the understating of complex and dynamic systems such as synapses.

The study presented here was well executed and discussed in the light of experimental data. The conclusions about the possibility of cooperative effects on receptor binding are remarkably interesting and valuable for the community.

However, I have some concerns about some of the values chosen for the simulation parameters:

- I think that there is a mistake in the number of slots P of table 1 (70) because after the simulation of LTP the number of bound AMPARs is more than 200 (Fig. 2). If I understood well, the number of slots is not changed during the simulation of E-LTP, and there are always slots in excess with respect to the number of bound receptors.

- Again regarding P values, they refer to the total number of slots of the synapse or the number of slots of a nanodomain? Several studies have revealed the presence of at least 2-3 nanodomains 70 nm-wide in a subset of synapses, each of these nanoclusters containing ~20 receptors (i.e. Nair et al 2013) so a P value of 25 would only represent one nanodomain but not the entire PSD.

- Regarding the rates of AMPARs entering (kin) or exiting (kout) the spine, they are not changed even after AMPAR exocytosis, is this right? But could we expect that the increase in the number of membrane receptors in the spine will shift the equilibrium towards a higher kout?

- In the cooperative model, how the maximum possible number of neighbors was calculated? is it four as the scheme in Fig. 3A? Does this value come from experimental data?

- Why the units of the binding rate are µm2/s? An explanation of the meaning of this value would help the reader to compare it with values obtained or employed in other studies.

Regarding the discussion:

- Do the authors think that there should be a minimum size for the synapse to undergo an efficient LTP? Indeed, a sigmoid relationship between bound AMPARs/mobile AMPARs is clearly observed only for large synapses. Could this mean that synapses with nanodomains are less efficient to keep receptors inside?

- The point 3 of the model predictions (lines 316-317) predicts that “receptors are not uniformly distributed in nanodomains… but cluster…”. Could the authors define what is a nanodomain and a cluster for them?

Minor :

- The last sentence in the author summary: “Characterizing the principles… the role of synaptic dynamics in neurodegenerative diseases” is rather out of context, as no other reference exists in the manuscript about these pathologies.

- Synapses are simulated as occurring in “spines”, what is actually the case of excitatory synapses in principal cells. However, the structure “spine” is not described in the introduction. Fig. 2 shows the structure without any explanation. The role of cytoskeleton in shaping the spine should also be mentioned.

- The representations of dwell time in Fig. 6C are difficult to interpret. Which are the units of “Position”?

- A typo: in line 318, replace “too” by “to”.

- Line 390: “GluR1” subunit is named “GluA1” in the current classification of AMPA receptors.

Reviewer #2: The MS by Becker and Tetzlaff presents a model of synapse function during synaptic plasticity centered on the role of various aspects of the trafficking and stabilization of the main category of glutamate receptors involved in synaptic transmission - i.e. AMPA receptors (AMPAR). Based on a solid and thorough analysis of the experimental data available on the topic, the authors model the contribution of AMPAR exocytosis and endocytosis, synapse growth, AMPAR diffusion and trapping on scaffold protein binding sites, including its potential cooperativity.

The authors aim at understanding the potential contribution of these various phenomenon to the “early” phase of activity dependent synaptic potentiation (eLTP) that is observed experimentally and is independent of receptor synthesis. In their model, the authors vary the rates of these various AMPAR trafficking routes and binding affinities, comparing simple and cooperative binding. They compare their simulations

The main conclusion of their study is that no single pathway can explain the experimentally observed eLTP but that a combination of increased AMPAR exocytosis and cooperative trapping can reproduce the experimental observations.

Altogether this is a solid and thorough study of the potential contribution of various AMPAR trafficking pathways to synaptic potentiation. It is relatively standard in its approach, but has the merit of sticking to experimental data. It uses state of the art knowledge of AMPAR receptor trafficking, although exclusively focusing on non-signaling pathways, and this is probably the biggest weakness of the study. Indeed, the wealth of data reporting altered AMPAR trapping during LTP for example is ignored. While the authors are correct in stating that early signaling events such as calcium rises or CaMKII activation is only transient, AMPAR modifications could very well be long lasting and this should be taken into account and introduced as a parameter.

Another point that is not explored, and should be, relates to the location of AMPAR exocytosis that is hotly debated and could be either in the spine or in the dendrite. It would be interesting to know the impact of this parameter on their model.

I did not fully get the interest of their structural LTP modelling as it seems to boil down to a change in AMPAR exocytosis rate and exocytosis content. This should be further clarified.

Minor points:

Figure 5: reference error to a non existing panel E

This statement l346 is unclear “We, therefore, predict that cutting of the PSD from the mobile pool would result in a rapid decay of excitatory synaptic transmission.”

I do not understand how the authors can make the following statement l366: “Furthermore, our results indicate that sLTP accompanied by increased AMPAR tracking from REs during E-LTP”

**Have all data underlying the figures and results presented in the manuscript been provided?**

Reviewer #1: Yes

Reviewer #2: Yes

PLOS authors have the option to publish the peer review history of their article (what does this mean?). If published, this will include your full peer review and any attached files.

Reviewer #1: **Yes: **Marianne Renner

Reviewer #2: No
---

## [Decision Letter · Decision Letter 1]

17 Feb 2021

Dear Mr. Becker,

We are pleased to inform you that your manuscript 'The biophysical basis underlying the maintenance of early phase long-term potentiation' has been provisionally accepted for publication in PLOS Computational Biology.

Best regards,

Joanna Jędrzejewska-Szmek, Ph.D.

Guest Editor

PLOS Computational Biology

Kim Blackwell

Deputy Editor

PLOS Computational Biology

Reviewer's Responses to Questions

**Comments to the Authors:**

Reviewer #1: The authors adressed all my queries and comments. Nice work!

Reviewer #2: The authors have adequately revised the MS that is now much improved and clearer. I particularly liked the addition of the extension of the model with a second receptor type which brings additional understanding.

My requests for clarifications are met

**Have all data underlying the figures and results presented in the manuscript been provided?**

Reviewer #1: Yes

Reviewer #2: Yes

PLOS authors have the option to publish the peer review history of their article (what does this mean?). If published, this will include your full peer review and any attached files.

Reviewer #1: **Yes: **Marianne Renner

Reviewer #2: No

---

## [Editor Report · Acceptance letter]

17 Mar 2021

PCOMPBIOL-D-20-02048R1 

The biophysical basis underlying the maintenance of early phase long-term potentiation

Dear Dr Becker,

I am pleased to inform you that your manuscript has been formally accepted for publication in PLOS Computational Biology. Your manuscript is now with our production department and you will be notified of the publication date in due course.

With kind regards,

Alice Ellingham
